# Exploring Inner Ear and Brain Connectivity through Perilymph Sampling for Early Detection of Neurological Diseases: A Provocative Proposal

**DOI:** 10.3390/brainsci14060621

**Published:** 2024-06-20

**Authors:** Arianna Di Stadio, Massimo Ralli, Diego Kaski, Nehzat Koohi, Federico Maria Gioacchini, Jeffrey W. Kysar, Anil K. Lalwani, Athanasia Warnecke, Evanthia Bernitsas

**Affiliations:** 1Department GF Ingrassia, University of Catania, 95131 Catania, Italy; 2Sense Research Unit, Department of Clinical and Movement Neurosciences, UCL Queen Square Institute of Neurology, London WC1N 3BG, UK; d.kaski@ucl.ac.uk (D.K.); n.koohi@ucl.ac.uk (N.K.); 3Organ of Sense Department, University La Sapienza, 00185 Rome, Italy; massimo.ralli@uniroma1.it; 4Ear, Nose, and Throat Unit, Department of Clinical and Molecular Sciences, Polytechnic University of Marche, 60020 Ancona, Italy; giox83@hotmail.com; 5Otolaryngology—Head and Neck Department, Columbia University, New York, NY 10032, USA; jk2079@columbia.edu (J.W.K.); akl2144@cumc.columbia.edu (A.K.L.); 6Department of Otolaryngology—Head and Neck Surgery, Hannover Medical School, 30625 Hannover, Germany; warnecke.athanasia@mh-hannover.de; 7Multiple Sclerosis Center, Neurology Department, Wayne State University, Detroit, MI 48201, USA; ebernits@med.wayne.edu

**Keywords:** brain, neuroinflammation, neurodegeneration, CSF, perilymph

## Abstract

Recent evidence shows that it is possible to identify the elements responsible for sensorineural hearing loss, such as pro-inflammatory cytokines and macrophages, by performing perilymph sampling. However, current studies have only focused on the diagnosis of such as otologic conditions. Hearing loss is a feature of certain neuroinflammatory disorders such as multiple sclerosis, and sensorineural hearing loss (SNHL) is widely detected in Alzheimer’s disease. Although the environment of the inner ear is highly regulated, there are several communication pathways between the perilymph of the inner ear and cerebrospinal fluid (CSF). Thus, examination of the perilymph may help understand the mechanism behind the hearing loss observed in certain neuroinflammatory and neurodegenerative diseases. Herein, we review the constituents of CSF and perilymph, the anatomy of the inner ear and its connection with the brain. Then, we discuss the relevance of perilymph sampling in neurology. Currently, perilymph sampling is only performed during surgical procedures, but we hypothesize a simplified and low-invasive technique that could allow sampling in a clinical setting with the same ease as performing an intratympanic injection under direct visual check. The use of this modified technique could allow for perilymph sampling in people with hearing loss and neuroinflammatory/neurodegenerative disorders and clarify the relationship between these conditions; in fact, by measuring the concentration of neuroinflammatory and/or neurodegenerative biomarkers and those typically expressed in the inner ear in aging SNHL, it could be possible to understand if SNHL is caused by aging or neuroinflammation.

## 1. Introduction

Neuroinflammation is a process implicated in several neuroinflammatory and neurodegenerative diseases. The term “neuroinflammation” is a broad definition to summarize a complex process that occurs within the brain [1]. DiSabato defined neuroinflammation as an inflammatory response within the brain or spinal cord, mediated by the production of cytokines, chemokines, reactive oxygen species, and secondary messengers. Neuroinflammation can be caused and triggered by systemic disorders (e.g., hypertension), infections (e.g., COVID-19), and autoimmune disorders (e.g., multiple sclerosis) [2].

Central to this inflammatory cascade, are the microglia, a cellular element belonging to the macrophage family [2]. These cells are extremely dynamic and can easily change their state. Paolicelli et al. in 2022, updated the nomenclature of these cells based on recent studies that had allowed identification of the following aspects of microglia: morphology, epigenetic, transcriptomic, proteomic, and metabolomic [3]. Disease-associated microglia (DAMs) have been identified in neuroinflammatory and neurodegenerative diseases. Microglia have been identified in the inner ear of patients with Meniere’s disease [4]. However, the role of these cells in the inner ear was not clarified and only a mechanism about inflammation was proposed; moreover, the researchers did not explain how these cells, typically identified in the brain, can be found in the inner ear. Recently, a systematic review conducted by an international team focused attention on the possible routes of exchange of cellular elements and inflammatory cytokines between the brain and inner ear [5]. The theoretical hypothesis was based on the exchange of fluids between cerebrospinal fluid (CSF) and perilymph. The authors illustrated the route of fluid exchanges and gave a mechanistic explanation, based on a systematic review of studies that focused on the investigation of auditory functions in patients with neuroinflammatory and neurodegenerative diseases, that the passage of pro-inflammatory elements could be responsible for hearing loss due to damage to the hair cells. This review aimed to identify the role of the inner ear in brain decline; currently, we know only that there is a coexistence of hearing loss and brain degeneration but not the actual link between these two conditions [6].

Warnecke et al. analyzed the perilymph content of patients with hearing loss and identified both inflammatory elements [7] and microRNA [8]. In their work, the authors stated that only very small microvolumes of perilymph were necessary to correctly perform the analyses and obtain relevant information [9].

Because perilymph shares some elements with CSF, the analysis of its contents could be useful to identify some typical elements of brain neuroinflammation/neurodegeneration.

Currently, perilymph sampling is only focused on the detection of inner ear diseases, and the potential of perilymph analyses to detect neuroinflammatory diseases in the early phase remains a novel concept. Furthermore, the complex interactions between neuroinflammation, neurodegeneration, and auditory dysfunction have not been explored.

Biomarkers for neuroinflammatory [10] and neurodegenerative disease [11,12] that have been identified in the CSF could also be identified in the perilymph [13]. In fact, one of the drawbacks of perilymph sampling is the risk of contamination with CSF [13]; when collecting perilymph samples in excess of 10 microliters, there is an increased risk of CSF contamination, which may lead to a lack of specificity in the resulting perilymph analyses [13]. This inherent limitation of the technique could be a unique opportunity to understand the relationship between neurological disorders and changes to auditory function, potentially signaling significant advancements in neurological research.

The objective of this review is to clarify how neuroinflammatory and neurodegenerative biomarkers are identified in the perilymph, to clarify their role in the brain–ear connection, and to define the role of neuroinflammatory elements in causing auditory alterations in patients with neurological disorders. A small sample of perilymph can be collected without damaging the inner ear cells and without the risk of causing the onset or worsening of hearing loss [14]. Herein, we discuss why and how to perform perilymph sampling in patients with neuroinflammatory and neurodegenerative diseases. In the subsequent sections, we will discuss the benefits of microneedle-assisted perilymph extraction technologies to understand the real relationship between neuroinflammatory and neurodegenerative diseases and hearing loss, with an emphasis on its potential to enable early diagnosis and treatment during potentially reversible stages of disease progression.

## 2. Cerebrospinal Fluid

CSF is an ultrafiltrate of plasma distributed between the subarachnoid space (125 mL) and the ventricles (25 mL) [15]. It provides brain nourishment and protection and is responsible for the removal of waste substances [16]. Given the absence of a significant barrier between the CSF and the extracellular space of the brain (ECSB), the blood–CSF barrier is instrumental to regulate the brain’s environment. The CSF allows only the passage of very small molecules, such as vitamins, and provides a barrier against large molecules [17]. Nourishment from blood is mediated via the axis choroid plexus (CB)-CSF-ECSB [15,16,18]. CSF removes waste produced by brain metabolism, such as peroxidation products, glycosylated proteins, excess neurotransmitters, debris from the lining of the ventricles, bacteria, viruses, and all unnecessary molecules [18]. The accumulation of some of these molecules, which is observed in aging and neurodegenerative diseases, is toxic to the brain and responsible for neuroinflammation/neurodegeneration [1,2]. CSF is important in supporting brain health.

In the case of neurodegenerative and neuroinflammatory diseases, several biomarkers can be identified in the CSF (Table 1); it is widely accepted that biochemical changes taking place in the brain are reflected in the CSF [19].

Research is ongoing for all biomarkers of both neurodegenerative and neuroinflammatory conditions. Current investigations are centered on identifying biomarkers, such as eotaxin-1 (CCL11) [21], that can predict the progression of the diseases and their development by examining light neurofilaments and some cytokines [22]. The latter, in particular CSF TNF-α, IL-10, CXCL13, and NF-L levels, can be associated with the development of MS [22].

## 3. Perilymph

Perilymph is an extracellular fluid located within the inner ear, found within the scala tympani and scala vestibuli of the cochlea. Sodium represents the major cation within perilymph, with concentrations of 138 mM, while potassium is present at a concentration of 6.9 mM [23,24]. Calcium is also present within perilymph [24,25]. The composition of this fluid closely resembles that of CSF and plasma [23,24,25,26]. Similar to CSF, perilymph is an active part of the perilymph–blood barrier (PBB), allowing the nourishment of the inner ear [26].

The contents of perilymph reflect the inner ear’s condition, in health and disease [7,8,9,14] (Table 2), much like CSF.

## 4. The Inner Ear

The human inner ear, located in the temporal bone, contains the cochlea (auditory organ), and the vestibular system. The external bony labyrinth contains the membranous labyrinth internally. Perilymph fills the space between the bony labyrinth and membranous labyrinth, while the membranous labyrinth contains endolymph, a unique extracellular solution characterized by high K^+^ and low Ca^2+^ and Na^+^ concentrations.

The cochlea, which runs around the modiolus, is divided into three different segments: basal, middle, and apical turns. These segments contain the same structures and the organ of Corti (scala media), which is limited superiorly by the scala vestibuli and inferiorly by the scala tympani. The latter are filled with perilymph. In contrast, the scala media, connected to the vestibular system via the ductus reuniens, contains the endolymph [26].

The cochlea has two windows: the oval window, which connects the inner ear and the middle ear, and the round window (RW). The stapes induces movement of the perilymph in the scala vestibuli, with the movement transmitted from the scala vestibuli to the scala tympani, where the RW is located [26]. Both windows are covered by small membranes that allow the normal movements of the perilymph [26]. Perilymph movement is fundamental in the auditory process; the fluid movements on the top and bottom of the organ of Corti stimulate the hair cells, initiating the auditory process [34]. The cochlear aqueduct, an essential anatomical channel, links the scala tympani to the subarachnoid, connecting the cochlear environment with the CSF-containing spaces [35,36].

## 5. The Potential Bridging Role of Perilymph in Future Neuro-Otological Practice

The inner ear and brain are connected through the cochlear aqueduct and internal auditory canal (IAC) . The model of fluid dynamics, shown in Figure 1, suggests that the anatomical connection provided by the cochlear aqueduct allows for the contents of the CSF to be present within the cochlea’s perilymph.

It is known that brain infections such as meningitis can cause permanent sensorineural hearing loss (SNHL) [37], and the levels of glucose (low) and proteins (high) in the CSF correlate with damage to the cochlea [38]. It has been shown that autoimmune anti-NMDA receptor encephalitis can cause cochlear damage, leading to bilateral hearing loss [39]. Other systemic conditions can cause damage to the inner ear through different mechanisms, including deposition of cytotoxic molecules [40].

Because inflammatory elements can pass from the brain to the inner ear through fluid exchanges, it is expected that similar findings will be found in both ears, regardless of the presence or severity of SNHL [4,37,38]. However, clinical research has shown that one ear can be more susceptible than the other, explaining the asymmetry of SNHL even in conditions where it would be expected to be symmetric [41]. This asymmetry might be related to a different immune response to the inflammatory elements between the two ears [42], with consequent optimal response to the inflammatory event on one side rather than the other.

It is also important to consider that the cochlear aqueduct is not always present [4], whereas the IAC is always present in humans; the absence of the cochlear aqueduct might limit fluid exchanges and, consequently, the inflammation caused by pro-inflammatory cytokines [4]. In these cases, despite the presence of inflammatory substances in CSF and potentially the perilymph, the concentration of these components will not be high enough to cause severe damage to hair cells with auditory impairment [43,44,45].

Perilymph sampling performed on a single side might be sufficient to obtain information about its contents, providing information and limiting the risk of performing bilateral sampling. In patients with asymmetric SNHL, the sample can be extracted from the ear with the worse threshold. In other cases, the choice of side will be made based on the more visible and accessible round window.

## 6. Risk, Limitations, and Solution for Perilymph Sampling

Because perilymph is fundamental in the auditory process, excessive sampling can expose the patient to immediate or delayed SNHL. The method of sampling and the quantity of perilymph are two important aspects to consider for safe sampling.

Most experiences of perilymph sampling in humans have been in patients affected by SNHL or during cochlear implant surgery or stapedectomy [14]; in both types of surgery, the reduction of perilymph quantity did not affect auditory results [9,14] because perilymph is rapidly replaced after a few weeks. Based on cochlear microphonics studies, it has been shown that loss of 5–10 microliters of perilymph is unlikely to cause SNHL [14,46].

Several studies have shown that a volume as small as 5 microliters of perilymph can provide a range of information, including the possibility of studying metabolomic, proteomic, extracting microRNA, and identifying inflammatory elements [14,33,46]. Other than its usefulness to detect markers of neuroinflammation and neurodegeneration, five microliters can be easily restored with no adverse outcome.

Another limitation of the technique is the risk of opening the RW membrane, which, if not properly sealed, could cause further leakage of perilymph, leading to an increase in the risk of infection. This is not a problem during surgery but would be relevant in clinical perilymph sampling. Several researchers have worked in this field to identify minimally invasive methods to open the RW membrane [46,47]. Advancements in microneedle technology [47,48], which can also be reproduced using 3D printing systems [49], enable multiple safe perforations of the RW membrane and ensure effective healing of both the RW membrane and tympanic membrane (TM) in the case of the intratympanic approach [49,50].

Finally, access to the RW through the external auditory canal can be challenging due to its position (median angle of 113°) and anatomical differences in the bones between individuals [51] that may require bone drilling or the use of an angulated flexible needle [14] to reach the RW membrane. The use of an endoscope with a direct view of the TM combined with an angulated flexible microneedle could be a solution for perilymph sampling.

## 7. Procedures and Techniques for Perilymph Sampling via a Microneedle

In 2016, Watanabe et al. proposed the use of a dual wedge microneedle [52] because it allows for small perforations of the RW membrane, facilitates aspiration, and provides precise volume control. The devised prototype was a wedge-shaped needle with a tip curvature of 4.5 μm and a surface roughness of 3.66 μm in root-mean-square. The needle can create an oval perforation with minor and major diameters of 143 and 344 μm, respectively, and allows for sampling of the perilymph in around 3 s. Further improvement to the microneedle was made by the same team in 2020 [53]. The authors confirmed the validity of their device both in terms of safety and minimal invasiveness in several animal studies [48,49,50]. Additionally, this microneedle is available in metallic [53] and glassy carbon [54] versions.

In 2019, Early et al. proposed a novel, minimally invasive microneedle device capable of reliably collecting a perilymph sample of at least 1 mL in volume with minimal contamination from middle ear fluids [47]. The needle has three different cuts that allow monitoring of the depth of penetration through the round window membrane and protection of intracochlear structures. The device creates a half-moon shape perforation, consistent with penetration of the beveled needle tip but stopping before the main needle shaft (Figure 2).

The authors performed their tests both in experimental settings and on human temporal bone and were able to extract 3.5 microliters of pure perilymph using this microneedle.

Recently, St Peter et al. developed a prototype for performing perilymph sampling [14]. The device has two internal actuators: one advancing a needle and one allowing a plunger to be withdrawn from the needle/internal reservoir. This allows for advancement of a needle from the curved tip of the device in submillimeter increments and withdrawal of up to 10 microliters of fluid. The tip of the device measures 0.86 mm at the tip from which the needle is deployed. Given the above, the potential of this device is noteworthy.

The microneedle technique facilitates perilymph sampling; however a clear visualization of the TM with its structure is fundamental for the safe execution of this procedure.

The advancement of endoscope-assisted ear surgery has introduced the use of rigid endoscopes with diameters less than 4 mm [55]. These endoscopes are used to directly visualize the middle ear and perform reconstructive surgery of the ossicular chain or the removal of cholesteatoma [55], and to insert cochlear implants through the RW [56] or the removal of schwannoma [57]. However, in these cases, the TM is lifted to introduce the fibroscope. Evidently, this is not a feasible option for performing a diagnostic procedure like perilymph sampling.

The endoscope in perilymph sampling should be “an assistant”. The micro-endoscope will allow a clear view of the TM and of the RW to collect perilymph with direct visual identification. In this way, it would be possible to pierce the TM and insert the microneedle into the RW (Figure 3).

Another option could be making a small hole, similar to tympanocentesis, and inserting a semi-rigid operative micro-endoscope (0.9 mm diameter) [58] to directly visualize the RW within the middle ear. Given this endoscope has an operative capability, it would be possible to insert a retractable microneedle into the endoscope and then, once close to the RW membrane, extract the needle for sampling. This suggested technique has even been used successfully to inject drugs into the inner ear in animals with preservation of the middle ear structures [59].

## 8. Discussion

Perilymph sampling can be a valuable tool for neurologists and neuroscientists. It can be easily used in the majority of patients, especially in those for whom a lumbar puncture can be problematic, such as people who are overweight or have a challenging lower back anatomy.

Due to the exchange of cellular and non-cellular elements between CSF and perilymph [5], sampling of perilymph could provide information about the relationship between neuroinflammatory [41,43] and neurodegenerative [6,59] disorders and SNHL.

It is known that deposition of reactive oxygen species (ROS) in the inner ear causes activation of macrophages and glial cells, increases the expression of pro-inflammatory cytokines and chemokines, and causes alterations in connexin (Cxs) and pannexin (Panx) expression, likely responsible for dysregulation of the microglia/astrocyte network. These events can damage the auditory pathway, both at the peripheral (ear) and central level (brainstem and cortex) [60], resulting in SNHL. Moreover, it has been shown that the mutation of NLRP3 induces peripheral (ear) SNHL [61]. The NLRP3 inflammasome is also a key element in Alzheimer’s disease (AD); its activation induces increased synthesis of pro-IL-1β and pro-IL-18 and activates caspase-1. The proteolytic caspase-1 processes the inactive IL pro-forms into their active pro-inflammatory forms. Once secreted, IL-1β and IL-18 promote the pyroptotic death of neurons [62]. Because NLRP3 is activated in AD by amyloid-beta (Aβ) [63], the identification of even small concentrations of Aβ in the perilymph could define a cause–effect link between brain neurodegeneration and SNHL.

Human studies that compared perilymph from healthy patients and patients with vestibular schwannoma found specific elements of the tumor such as TNF-α [7], a pro-inflammatory cytokine, notably accountable for SNHL in these tumors. TNF-α causes SNHL through the activation of NLRP3 [64]. TNF-α is a strong predictor of MS relapsing events [60], so based on its aforementioned effect, the identification of high concentrations of this element in the perilymph could, like Aβ, determine a causal link between neuroinflammatory disorders and SNHL [63].

Furthermore, the identification of microglia cells in the perilymph, which are present in both neuroinflammatory [22] and neurodegenerative conditions [65], might present a direct link between neuroinflammatory diseases and SNHL. As recently shown in a human temporal bone study [66], microglia cells can damage both the organ of Corti and the cochlear nerve; damage to both structures causes SNHL, respectively, cochlear and neural/central hearing impairment.

In summary, the identification of biomarkers related to neuroinflammatory and neurodegenerative disorders in the inner ear (perilymph) in patients with SNHL can define a causal link between neurological disorders and SNHL, rather than a simple association [6]. In the near future, once these relationships have been confirmed, it could be possible to monitor and evaluate the efficacy of treatments by simply evaluating auditory functions; in fact, the inhibition of NLRP3—mentioned as a potentially key factor to stop neurodegeneration [62]—might reverse SNHL in the early stage and stop its progression in the moderate degree too, so a pure-tone audiometry test associated with distortion product otoacoustic emissions (DPOAEs) might be an easy way to monitor drug efficacy.

## 9. Conclusions

Perilymph sampling performed in clinical settings might be useful to determine the correlation between neuroinflammatory/neurodegenerative diseases and SNHL, and to monitor the progression of neurological disorders and the efficacy of treatments used to revert/stop the neuroinflammatory process. Performing perilymph sampling using a micro endoscope—for direct view—microneedle—for reduced invasiveness—and microscopic device—for minimal and well-controlled sampling, might make the procedure feasible and practicable even in clinical settings, just like performing intratympanic steroid injections.

## Figures and Tables

**Figure 1 brainsci-14-00621-f001:**
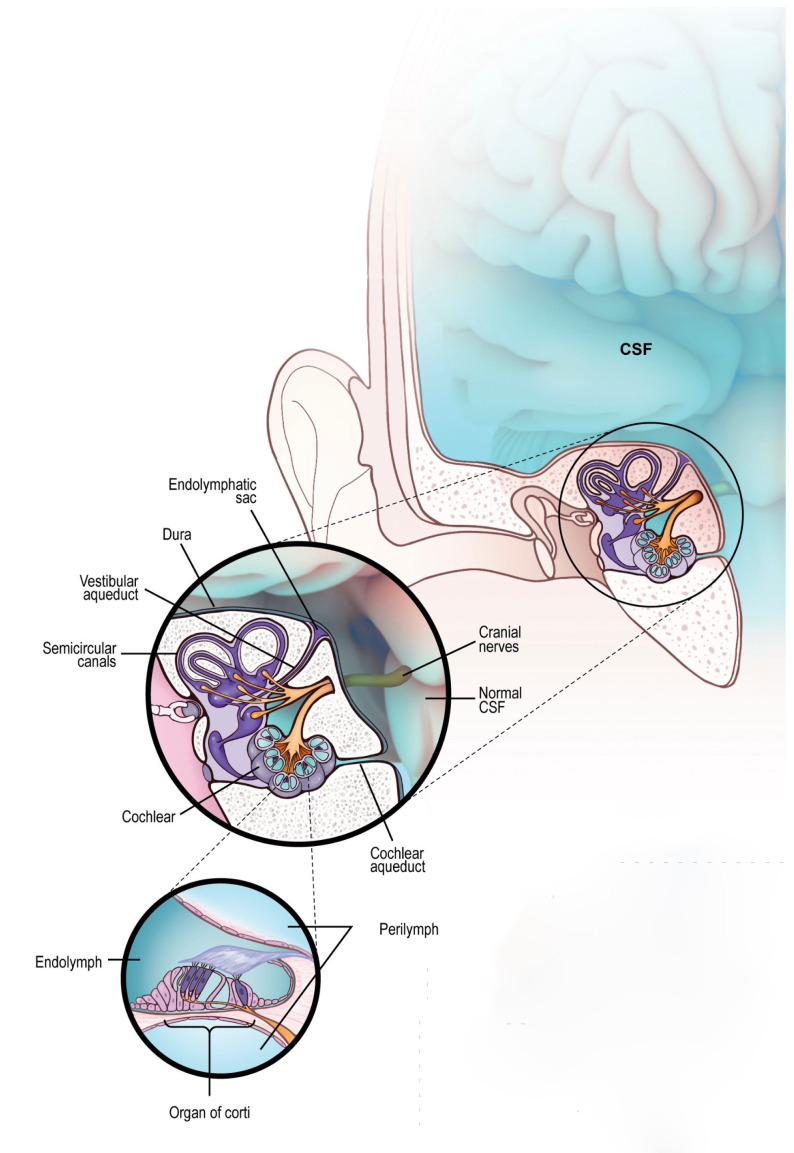
Liquid exchange between brain and inner ear. CSF (in blue) contained in the brain can normally pass into the inner ear via the cochlear aqueduct or through the internal auditory canal reaching the cochlea, sharing its content with the perilymph. Even the perilymph can retrogradely share some elements with CSF, although the exchange is limited by the small amount of the liquid contained into the inner ear. This is a physiologic mechanism of fluid mobilization between these two organs.

**Figure 2 brainsci-14-00621-f002:**
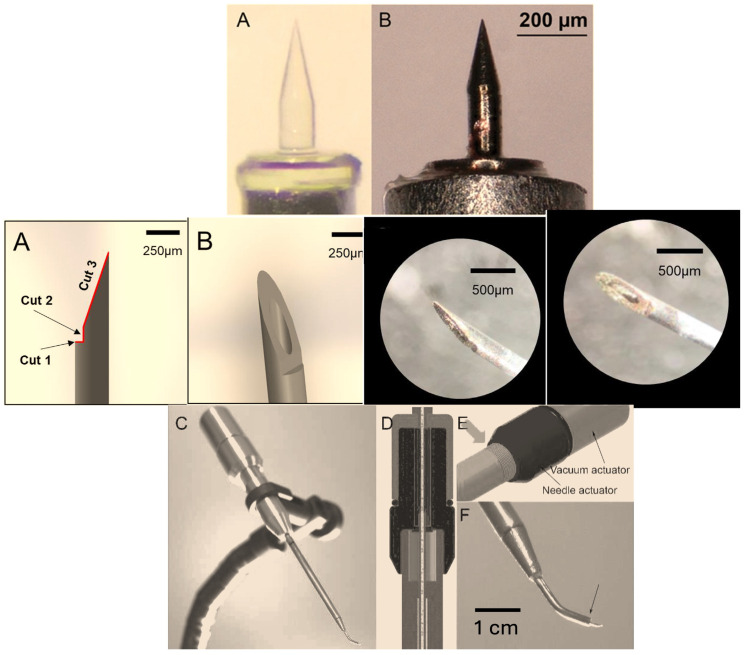
The image shows the microneedle proposed by Watanabe et al. (**top** (**A**,**B**)) [52], the one proposed by Early et al. (**middle**), scale bar = 250 µm (**A**,**B**) [47], and the prototype proposed by St Peter et al. [14] (modified figure from [14] JCM of MDPI (**bottom** (**C**–**F**)).

**Figure 3 brainsci-14-00621-f003:**
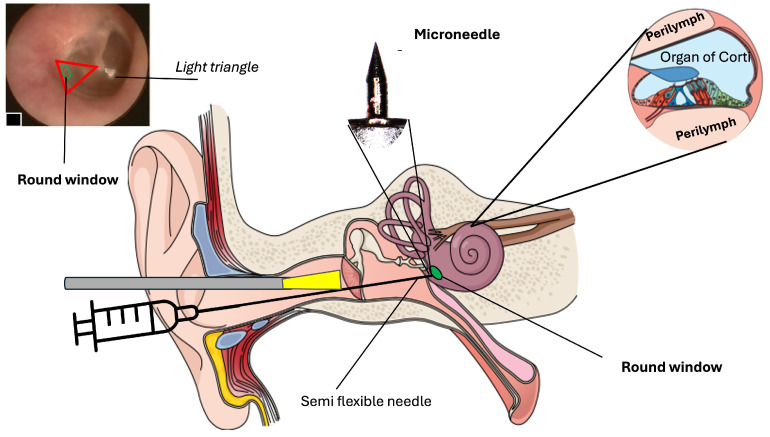
The scheme illustrates the technique of perilymph sampling under an endoscope guide. The red triangle area delimits the space where it is possible to identify the round wound niche (green circle). Once the round window (RW) has been identified, it is possible to go through the tympanic membrane (TM) using the device proposed by St. Peter et al. [14] The microneedle will come out from the device only after perforation of the TM just to reach the RW and perform the perilymph sampling.

**Table 1 brainsci-14-00621-t001:** Summary of the biomarkers of neurodegenerative (Alzheimer’s disease, AD) and neuroinflammatory (multiple sclerosis, MS) disorders [20,21,22] that can be identified in the CSF.

CSF Biomarkers	MS	AD
TNF-α	x	
IL-10	x	
IL-1β	x	x
NF-L	x	x
IL-9	x	
Apolipoprotein C-I	x	
Apolipoprotein A-II	x	
Anti-NF-L antibodies	x	
Fibulin 1	x	
A1AC	x	
A2MG	x	
IgG oligoclonal bands (OCB)	x	
k-FLC	x	
k-FLC index	x	
CXCL13	x	
YKL-40 (CHI3L1)	x	
GFAP	x	x
miR-142-3p	x	
MCP-1 (CCL2)		x
CXCL10		x
BACE1		x
Tau protein (t-tau)		x
Aβ40		x
Aβ42		x
TREM2		x

TNFα: tumor necrosis factor α; NF-L: neurofilament light chain; IL-10: interleukin 10; IL-1β: Interleukin 1 beta; IL-9: interleukin 9; A1AC: alpha-1 antichymotrypsin; A2MG: alpha-1 macroglobulin; CXCL13: chemokine ligand 13; CXCL10: chemokine ligand 10; GFAP: glial fibrillary acidic protein; k-FLC: kappa free light chains; BACE1: β-site APP-cleaving enzyme 1; TREM2: triggering receptor expressed on myeloid cells 2; Aβ40: amyloid beta, hyperphosphorylated isoform 40; Aβ42: amyloid beta, hyperphosphorylated isoform 42; MCP-1: monocyte chemoattractant protein 1; YKL-40: chitinase-3-like protein 1. “x” indicates the presence of the elements in the CSF.

**Table 2 brainsci-14-00621-t002:** Summary of the studies conducted on human perilymph.

	Years	Number of Subjects	Cause of Hearing Loss	Perilympatic Components Analyzed	Results
Mavel et al. [27]	2018	23	CMV; trauma; MD	Metabolome	A fingerprinting was obtained from 98 robust metabolites
Edvardsson Rasmussen et al. [28]	2018	16	VS	Proteome	Alpha-2-HS-glycoprotein, P02765, was shown to be an independent variable for tumor-associated hearing loss
Lin et al. [29]	2019	5	MD	Proteome	A total of 228 proteins were identified that were common across the samples from patients with Meniere’s disease, showing 38 proteins with significantly differential abundance
Thrin et al. [30]	2019	19	n/a	Metabolome	A total of 106 different metabolites were identified; metabolomic profiles were significantly different for subjects with ≤12 or >12 years of hearing loss
de Vires et al. [31]	2019	38	MD, CMV, EVA, CHARGE, meningitis	Proteome	(1) BDNF is expressed in cochleartissue in normal hearing individuals; (2) there was overall a decreased level of expression of BDNF-regulated proteins in profoundly hearing-impaired patients compared to patients with some residual hearing
Warnecke et al. [7]	2019	43	n/a	Proteome	Multiplex protein analyses are feasible in very small samples (1 microL or less); higher IGFBP1 levels were measured in patients with complete loss of auditory function compared to patients with residual hearing
Shew et al. [32]	2021	10	MD	miRNA	In the perilymph of patients with MD, authors identified 16 differentially expressed miRNAs
Schmitt et al. [33]	2021	31	MD, OS, EVA	Proteome	Overall, 895 different proteins were found in allsamples; based on quantification values, a disease-specific protein distributionin the perilymph was demonstrated
van Dieken et al. [34]	2022	38	n/a	Proteome	Authors proposed a human protein atlas of the cochlea

OS: otosclerosis; VS: vestibular schwannoma; CMV: citomegalovirus infection; MD: Meniere’s disease; EVA: enlarged vestibular aqueduct; BDNF: brain-derived neurotrophic factor; CHARGE: coloboma, heart defect, atresia choanae, retarded growth and development, genital hypoplasia, ear anomalies/deafness; n/a: not applicable.

## Data Availability

Data are available under request to the corresponding author.

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
