# Peer review of "Exploring Inner Ear and Brain Connectivity through Perilymph Sampling for Early Detection of Neurological Diseases: A Provocative Proposal"

_brainsci, 2024, doi:10.3390/brainsci14060621_

Round 1
Reviewer 1 Report
Comments and Suggestions for Authors
It is an increasing clinical interest to analyse perilymph regarding neuroinflammatory and neurodegenerative biomarkers. The aim is to clarify their role in brain-ear connection and their possible role in auditory alterations in patients with neurological disorders but also to detect biomarkers involved in development of sensorineural hearing loss.
The present review give a useful background to the components involved in the brain-ear connections and also summarize the methods described to obtain perilymph in patients subjected to neurological investigation without the onset and worsening of hearing loss.
The review, written by distinguished experts in this area, is titled “A provocative proposal.” However, investigating perilymph is of utmost importance and could not only describe interactions between neurological diseases and audiological influence but also shed light on our understanding of how neuroinflammation and aging are involved in the development of sensorineural hearing loss.
The review is well written and nicely illustrated. It may in fact also be an excellent chapter in a neurological textbook. The section on collecting of perilymph is carefully written without any suggestions which may harm the patient. The review will be a good background manual for those clinicians which are entering this new investigative area.
To summarize I definitely recommend this review for publication in its present form in Brain Sciences.
Author Response
Dear reviewer,
Thank you for the time you dedicated to review our article, the appreciation of our work and your positive feedback.
Reviewer 2 Report
Comments and Suggestions for Authors
Good research and significant information.
Minor concerns
1. M1 vs M2 is outdated. Please avoid referring to such simple dichotomy.
2. Please summarize and table not only the CSF biomarker but also the previous reports of markers in the inner ear.
Author Response
Dear reviewer,
thank you for the time dedicated to review our article and the appreciation of our work.
Regarding your concerns we addressed them as following
M1 vs M2 is outdated. Please avoid referring to such simple dichotomy.
We updated the terms following the recent updates about microglia “Microglia state and Nomenclature” published on Neuron in 2022.
- Please summarize and table not only the CSF biomarker but also the previous reports of markers in the inner ear.
We added a table (table 2) that summarizes the studies performed on the humans’ inner ear and we updated the references including these studies.
Reviewer 3 Report
Comments and Suggestions for Authors
Dear authors,
This review article is well-done to review the mehtod to detect the compnent of the perilymph in the inner ear. Due to connection between the perilymph in inner ear and CSF in the brain, the disease of the brain may be coexisted in the inner ear or share the same etiology such as aging or neuroinflammation.
I just have one comment about the role of endolymph which may be more difficult to collect and analyze but may present the main factors to cause the hearing loss. The authors seem not to mention about this.
Author Response
Dear Reviewer,
Thank you for your comments and the appreciation of our work.
Because we want to discuss a simple procedure that can be performed in clinical setting, the endolymph sampling, which as you correctly underlined is complex, was not mentioned. We are aware that the analysis of endolymph can offer important information, however, is too much complex to be performed in clinical setting, reason why we did not mention it in this review.
Moreover, the paper was prevalently focused on CSF and perilymph and their exchange of elements to clarify the relationship between neurologic diseases and hearing functions; the endolymph, which is produced by secretory cells in the stria vascularis of the cochlea and the dark cells of the vestibular labyrinth, can undoubtedly gives info about what’s happened into the inner ear- but exclusively in this organ.